# Preparation of Anti-Bacterial Cellulose Nanofibrils (CNFs) from Bamboo Pulp in a Reactable Citric Acid–Choline Chloride Deep Eutectic Solvent

**DOI:** 10.3390/polym15010148

**Published:** 2022-12-28

**Authors:** Yuanchen Zhu, Jinhui Zhang, Dawei Wang, Zhengjun Shi, Jing Yang, Haiyan Yang

**Affiliations:** Key Laboratory for Forest Resources Conservation and Utilization in the Southwest Mountains of China, Ministry of Education, College of Materials and Chemical Engineering, Southwest Forestry University, Kunming 650224, China

**Keywords:** cellulose nanofibrils, anti-bacterial, bamboo, deep eutectic solvent

## Abstract

In this study, bamboo pulp was simultaneously fibrillated and esterified in one-pot citric acid–choline chloride deep eutectic solvent treatment. The results indicated that increasing the temperature and time promoted esterification, yielding 0.19 to 0.35 mmol/g of the carboxyl group in CNFs. However, increasing the temperature and time resulted in decreases in yields and the diameter of CNFs from 84.5 to 66.6% and 12 to 4 nm, respectively. Analysis of the anti-bacterial activities of CNFs suggested that the high carboxyl group content corresponded to the effective inhibition of *Escherichia coli* and *Staphylococcus aureus* Taking yield, size, carboxyl group content, and anti-bacterial activate into consideration, treatment at 120 °C for 24 h was the optimal condition, yielding 76.0% CNF with 0.31 mmol/g carboxyl groups with a diameter of 8 nm and the inhibition fof *E. coli* (81.7%) and *S. aureus* (63.1%). In addition, effect of different CNFs on characteristics of polyvinyl alcohol (PVA) films were investigated. The results indicated that CNF obtained from the optimal condition was a favorable additive for the composite film, which enhanced (74%) the tensile strength of composite film compared with the pure PVA film due to its considerable size and carboxyl group content. However, the composite films did not show an anti-bacterial activate as CNF.

## 1. Introduction

Lignocellulosic biomass (LCB) is abundant in nature and is considered to be one of the important candidates for fossil resources due to its numerous attractive features [1]. Across the broad range of LCB, cellulose is a ubiquitous natural polymer, accounting for about 30–50% dry weight of LCB. Biodegradability, renewability, strong mechanical properties, low density, low cost, and environmental sustainability offer tremendous possibilities for the use of cellulose in various fields [1,2,3,4].

Cellulose is an assembly of glucose units. The cellulose chains bundle together through van der Waals and hydrogen bonds to form a three-dimensional micro-fibrillated structure and have a high stability against acids, harsh temperatures, and proteolytic enzymes [5]. Furthermore, cellulose coexists with other structural components in the LCB matrix, making it challenging for the isolation of cellulose [4]. Particularly, cellulose with a nanostructure has high thermal stability, high mechanical strength, and UV-blocking performance and has gained significant attention in the development of antibacterial biomaterials [6]. Thus, many methods have been applied to extract and disintegrate cellulose into a nanostructure.

Among the treatment technologies, chemical treatment could potentially destroy the hydrogen bonds in cellulose. A deep eutectic solvent (DES) containing a hydrogen bond accepter (HBA) and hydrogen bond donor (HBD) with multiple advantages (such as low vapor pressure and high tunability) is an alluring solvent for LCB fractionation to produce a nanomaterial. In the DES, HBA and HBD molecules can competitively form hydrogen bonds with cellulose and fibrillate into a nanostructure [7]. Taking organic acids as the HBD, the glycosidic bond of cellulose might be cleaved by the electron-withdrawing group of DES. Furthermore, the hydroxyl group of cellulose might be esterified by the acid in DES, yielding aa carboxyl-modified nanocellulose material [8]. The carboxyl-modified nanocellulose is a potent antibacterial agent and can be used as a carrier for antibacterial agents [9].

In this study, bamboo pulp (BP) was fibrillated and modified by a reactable choline chloride-citric acid deep eutectic solvent in a one-pot treatment. Yield, degree of substitution, zeta potential, diameter, and anti-bacterial effect of CNFs were investigated by titration, microscopic electrophoresis device, transmission electron microscope (TEM), and succussion, respectively. The effect of CNFs on the characteristics of polyvinyl alcohol films were analyzed by a UV–Vis spectrophotometer, thermogravimetric analysis, and a mechanical testing machine.

## 2. Materials and Methods

### 2.1. Materials

Bleached bamboo kraft pulp (BP) containing about 83.8% glucan, 6.5% xylan, and 1.3% lignin was purchased from Yunnan Yunjing Forestry and Paper Industry Ltd., Jinggu, China. Choline chloride (ChCl) and anhydrous citric acid (CA) were purchased from Sigma-Aldrich Co., Ltd., St. Louis, MO, USA. Polyvinyl alcohol (PVA) and NaOH were received from Shanghai Chenqi Chemical Technology Co., Ltd., Shanghai, China.

### 2.2. Preparation of CNFs and Composite Films

DES was prepared by mixing ChCl and CA at a molar ratio of 1:1 at 60 °C under stirring until a uniform solution was obtained. After cooling to room temperature, the DES was stored in a desiccator for further use. Fibrillation and esterification of BP was conducted as mixing 2.5 g BP with 50 g DES under stirring at 100 °C, 120 °C, and 140 °C for 12 and 24 h, respectively. At the end of the reaction, 50 mL water was used as an anti-solvent and added into the solution to quench the reaction. The mixture was treated with a refiner at 10,000 r/min for 10 min and repeated thrice. Then, distilled water was added into the mixture and the pulp diluted to 1 wt% (based on weight of origin pulp). The pulp solution was subsequently submitted to colloidal milling and treated at 6000 r/min for 40 min. After that, the solution was dialyzed with distilled water to remove the residual DES. Then, the suspension solution without DES was concentrated and freeze dried to prepare CNFs. The yield CNF samples were labelled according to the pretreatment temperature and time as listed in Table 1.

CNF-PVA composite films were prepared by adding 1 g PVA into 10 mL distilled water, stirred continuously, and refluxed at 90 °C for 1 h to obtain a homogeneous PVA solution. After cooling to room temperature, 0.05 g CNF was added into the PVA solution. The mixture was stirred for 1 h and further ultrasonically treated for 5 min to eliminate bubbles for the film preparation. Films were obtained by pouring the solutions into the Petri dish and drying at 40 °C for 24 h. After drying, the films were stripped, labeled according to the type of CNF, and submitted to further analysis. In this study, the dosage of CNF was confirmed according to previous research [10,11].

### 2.3. Characterization of CNFs

Transmission electron microscope (TEM) characterization was performed on a JEM-2100 (JEOL, Tokyo, Japan) operated at 200 kV. Before TEM imaging, 5 μL 0.1% CNF suspension solution was dropped onto a carbon-coated electron microscope grid and negatively stained with 1% phosphotungstic acid solution to reinforce the images.

Zeta potential of the CNF suspension was recorded on a microscopic electrophoresis device at room temperature (Malvern Zeta sizer Nano ZS90, Malvern, UK). Before analysis, the suspension solution containing 0.1 wt% CNF at pH 6 was ultrasonicated at 45 kHz for 10 min. All measurements were carried out in triplicate with each measurement of 10 runs.

FTIR spectra of CNFs were recorded on a Nicolet FTIR spectrometer (Nicolet Is50, Thermo Fisher Scientific, Waltham, MA, USA) by the standard KBr pellet method. A total of 64 scans were conducted at a resolution of 4 cm^−1^ in the range of 4000–400 cm^−1^.

Crystallinity nature: The CNFs was determined by a Shimadzu-6100 (Tokyo, Japan) diffractometer with Cu Kα (λ = 0.15406 nm) radiation. All samples were scanned at a scanning rate of 5°/min at 40 kV in the range of 5–40°. The crystalline index and interplanar spacing of CNF were calculated on the software according to the literature as Equations (1) and (2), respectively [12]:(1)$$\mathrm{CrI}=\frac{I_{020}-I_{\mathrm{am}}}{I_{020}}$$
(2)$$\mathrm{Interplanar}\;\mathrm{spacing}=\frac{0.9\;\lambda }{\beta \;\cos\theta }$$
where I_020_ is the maximum intensity of peak at about 2θ = 22°, I_am_ is the minimum intensity of the peak at about 2θ = 18°, corresponding to the amorphous region of cellulose, λ is the X-ray wavelength, θ is the scattering angle, β is the half width of diffraction peak.

The carboxyl group content in CNFs was determined by a NaOH titration method according to [13]. In detail, 30 mL 1% CNF suspension, 0.5 mL NaCl (1 mM) and 0.5 mL HCl (0.1 M) were added into a beaker and stirred for 30 min. Afterward, the mixture was titrated with 0.01 M NaOH aqueous solution. The content of the carboxyl group was determined based on Equation (3):(3)$$C_{\left(-\mathrm{COOH}\right)}=\frac{\left(V_{2}-V_{1}\right)C}{w}$$
where V_1_ and V_2_ represent the volumes of NaOH (mL) solution at the beginning and end positions of the turning point in the conductimetric titration curves, C represents the concentration of NaOH solution (0.01 M), w represents the absolute dry weight of the CNF (g).

### 2.4. Characterization of Composite Films

The opacity of the composite films was measured on a UV–Vis spectrophotometer (PerkinElmer, Waltham, MA, USA). Rectangular samples 1 × 2.5 cm were prepared and placed in the spectrophotometer. Absorbance spectra were recorded between 200 nm and 800 nm.

Thermal stability of the composite films was recorded using thermogravimetric analysis (TGA) and differential thermal analysis (DTA) on a simultaneous thermal analyzer (TG209F3; NETZSCH, Selb, Germany). The apparatus was continually flushed with nitrogen and run from room temperature to 600 °C at a rate of 10 °C/min.

Mechanical properties of the composite films were determined by a universal mechanical testing machine (ETM Type D; Wance, Shenzhen, China). Composite films cut into rectangular slices with a length of 100 mm, a width of 15 mm, and an average thickness of 0.1 mm were used for all of the measurements. The gauge length was adjusted to 20 mm at a strain rate of 1 mm/min. All experiments were performed ten times, and the average value was given.

### 2.5. Anti-Bacterial Activate Determination of CNFs and Composite Films

*Staphylococcus aureus* (*S. aureus*, G+) and *Escherichia coli* (*E. coli*, G−) were used to investigate the anti-bacterial activate of CNFs according to pervious research [14]. The bacterial species were pre-incubated in 50 mL sterilized broth (3 g/L beef extract and 5 g/L peptone in water) at 37 °C under shaking for 24 h. After incubation, 0.03 mol/L phosphate buffer solution (PBS, pH = 7) was diluted to obtain 1 × 10^5^ colony-forming units (CFU)/mL bacteria solutions. Before the anti-bacterial activate analysis, CNF samples were exposed to UV radiation in a clean bench (SW-CJ-1F, AIRTECH, Suzhou, China) for 30 min. Then, 0.15 g BP (or each CNF) and 100 μL of bacteria suspension were added into 15 mL PBS and inoculated at 24 °C for 18 h. At the end of incubation, 1 mL of bacterial suspension was taken out and diluted by ten-fold dilution to plate on the agar plates. After 24 h of incubation, the number of live bacteria was counted. The no-sample-added group was used as the black. The inhibition of CNFs was calculated as in Equation (4):(4)$$I=\frac{\mathrm{Wt}-\mathrm{Qt}}{\mathrm{Wt}}\times 100\%$$
where Wt is the colony forming units of black; Qt is the colony forming units of broth containing BP or CNF. Anti-bacterial activities of the pure PVA and nanocomposite PVA films were tested according to the method mentioned in a previous work [10].

## 3. Results and Discussion

### 3.1. Structural Characteristics of CNFs

DES can break down the initial intermolecular hydrogen bonds and establish new hydrogen bonds between DES and hydroxyl groups of cellulose, further altering the crystalline structure of cellulose [15]. Furthermore, DES has an acidolysis ability, which can also cleave the linkages in cellulose units. Among the acidic DES, carboxylic acid-based DES has been found to generate carboxyl groups on a cellulose surface [8]. In addition, the hydrogen bond accepter (ChCl) can also act as a phase transfer catalyst, inhibit the ionization of carboxyl acids, and further promote esterification [8,16]. Table 1 suggests that the yields of CNFs varied in the range of 84.5% and 66.6% according to the increase in the temperature and duration. It is speculated that DES treatment eliminated the partial composition of BP. High temperature and long reaction time introduced the degradation of hemicelluloses and partial amorphous cellulose. These degradation products might redeposit on the CNF surface as an anti-solvent is added into the mixture, leading to a dark suspension of CNF5. This phenomenon has also been observed in the study of the treatment of cotton cellulose with choline chloride and oxalic acid dihydrate [17]. However, the light color of suspension CNF6 compared with CNF5 might be due to the fact that the degradation products were partly evaporated during a long duration. The surface hydroxyl bonds of cellulose were also esterified during DES treatment, increasing the content of carboxyl groups from 0.19 to 0.35 mmol/g. The rise in the carboxyl group content in CNF with an increase in temperature and duration indicated that an increase in temperature accelerates the penetration of DES molecules into BP and increases the activity of BP and DES, resulting in a high content of carboxyl groups. Meanwhile, prolonging the duration permitted sufficient reaction between DES and BP, which also contributed to the high carboxylic group contents of samples CNF4 and CNF6 compared to CNF3 and CNF5. The carboxyl groups in CNFs were observed in the FTIR spectra (Figure 1b). The increase in the carboxyl group contents was also associated with the increase in the zeta potential of CNFs. As the carboxyl group content increased from 0.19 to 0.35 mmol/g, the zeta potential of cellulose increased from −6.24 to −16.63 mV. These results confirm that the CNFs possessed negatively charged surfaces generated from the carboxylate groups after DES treatment. Furthermore, the increase in the zeta potential allows for better stability of the suspension due to the electrostatic repulsion between the nanometric particles [18].

Potential alterations in the structure of BP during DES treatment were detected by FTIR and are shown in Figure 1b. The broad peak between 3350 cm^−1^ corresponded to the stretching vibration of O–H, the in-plane bending vibration absorption of which was observed at 1320 cm^−1^. The peak at about 1730 cm^−1^ was attributed to the stretching vibration of C=O in the carboxylic groups. The peak intensity increased gradually after DES treatment, consisting of the increase in the carboxyl group content shown in Table 1. The peak at 1640 cm^−1^ was related to the antisymmetric stretching vibration absorption of C=O. Peaks at 1165, 1112, and 1050 cm^−1^ were attributed to the stretching vibration of C–O–C and C–O, respectively. The peak at 898 cm^−1^ was ascribed to the vibration absorption band of C–O–C in the *β*-glycosidic linkages. The decreasing peak intensity indicated broken glycosidic linkages, introducing a decrease in the crystallin index of CNFs. As shown in Figure 1c, peaks at about 16°, 22°, and 34.0° in the diffractogram of BP were assigned to the 001, 020, and 004 planes of cellulose, respectively. After DES treatment, the peak at 34.0° disappeared and the intensity of the peak at about 22.0° decreased. The crystalline index of CNFs decreased from 75.2 to 46.2 (Figure 1d). It was assumed that thee DES competed with the anions and cations in hydrogen bonds in cellulose and disrupted the hydrogen bonds of cellulose [19]. The carboxylic groups prevented the rearranging of cellulose chains due to electrostatic repulsion and steric hindrance as the DES was removed. The changes in the crystalline index consisted of the carboxyl group contents.

The morphological features of the CNFs were analyzed by TEM and are shown in Figure 1e. The TEM images showed that the CNFs exhibited a sharp, rod-like appearance. The diameter of the CNFs varied in the range of 4–12 nm. CNFs from 24 h treatment had a narrow average diameter distribution compared to that from the 12 h treatment. Particularly, treatment at a temperature of 140 °C for 24 h introduced a distinguished decrease in the diameter of CNF6. This result consisted of the degradation of cellulose, as shown in Table 1. The cleavage of the glycosidic bonds of cellulose has also been observed in the treatment of sugar beet pulp with choline chloride–lactic acid [20].

### 3.2. Opacity of Films

Dispersion of the nanofiber throughout the film may result in light scattering. The transmittance spectra of the pure PVA and CNF–PVA composite films are displayed in Figure 2. The peak at about 300 nm in Figure 2a might be ascribed to the n–π* transition of carboxyl groups [21,22], whose intensity increased with the carboxyl group content in CNFs. The pure PVA film had the most transparent 87.4%. Composites with CNFs decreased the transparency of the films due to the light scattering by CNF. Despite this decrease in transparency, the composite films were still clear enough to see through easily (Figure 2b). Compared with other CNFs, the CNF1–PVA composite film had high opacity. This phenomenon might be due to the fact that CNFs with high zeta potential corresponds to better stability and good dispersibility in PVA. The related homogeneous structure with transparency has also been found in nanocellulose–PVA composite films [23,24]. However, the UV-blocking properties of the CNF–PVA composite films was much lower than the citric acid–cysteine–PVA film. This might be due to the fact that citric acid and cysteine molecules crosslinked PVA chains through covalent bonds [22]. However, the CNFs were added as solid particles and dispersed in PVA, which can scatter the light.

### 3.3. Thermal Stabilities of Composite Films

The effect of CNFs on the thermal stability of the PVA film is shown in Figure 3a,b. There were three stages in the DTG curves of films. An 8% weight loss at about 60–120 °C was ascribed to the evaporation of the absorbed moisture. The main weight loss of the samples was observed in the second stage in the range of 250–350 °C. This stage was attributed to the pyrolysis of the main composition of the films. Composites with CNFs led to a slight shift in temperature for the maximum decomposition rate from 304 °C to 312 °C. The shift might be due to the hydrogen bonding between CNF and PVA. The third stage (370–600 °C) was attributed to finial decomposition to ash as well as the decomposition of cellulose into D-glucopyranose monomers and then into free radicals [25]. Thus, films contained CNFs had a higher decomposition rate in this range compared to pure PVA due to the decomposition of cellulose.

### 3.4. Mechanical Properties of Composite Films

Cellulose nanoparticles are reported to improve the mechanical properties of PVA due to cross-linking with PVA [26]. The effect of different CNFs on the stress, strain, and Young’s modulus of the composite films is shown in Figure 4. At a CNF concentration of 5%, the tensile strength of the films increased from 57 to about 90 MPa. Compared with CNFs obtained from 100 °C, CNFs with a high content of carboxylic groups showed a favorable enhancement in the tensile strength of the composite film. However, CNFs from a short duration with a larger diameter preferably reinforced the strain of films compared with the smaller CNFs from a longer duration treatment. This result suggests that larger size CNFs cross-linked better with PVA molecules. Taking both the diameter and zeta potential into account, CNF3 with considerable diameter (10 nm) and carboxylic groups (0.31 mmol/g) showed a favorable effect on the mechanical properties of the films, simultaneously enhancing the tensile strength to 75.4% and strain of 80.9%.

### 3.5. Anti-Bacterial Activate of CNFs and Composite Films

Bacteria is one of the main initiators to cause food spoilage. The inhibition of bacteria is of great significance for food packaging [27]. The bacteriostatic rates of the CNFs were calculated by counting the bacterial colonies. Samples with an anti-bacterial efficiency higher than 70% were considered to have an antibacterial effect [28]. Figure 5 depicts the anti-bacteria effect of BP and CNFs against *E. coli* and *S. aureus*. The inhibition of the samples demonstrated that BP had no obvious anti-bacteria efficiency. DES treatment endowed an anti-bacterial effect on the CNFs. CNFs obtained from 100 °C and 120 °C treatment (both 12 and 24 h duration) had a favorable anti-bacterial effect compared to BP. However, these samples exhibited a different efficiency for *E. coli* compared to *S. aureus*. Significant inhibition was observed in the CNFs from 140 °C treatment for 24 h, which nearly approached the eradication of both bacterial strains (91.7% for *E. coli* and 96.4% for *S. aureus*, respectively). The positive correlations between the anti-bacterial efficiency and carboxyl group contents of CNF were observed, which was fitted in linear. The carboxyl group was also corelated with the content of the citric acid side chain on the cellulose backbone. The good linear fit of correlation between the citric acid contents and *E. coli* inhibition efficiency might be ascribed to the fact that citric acid introduced the inactivation of superoxide dismutase and thus inhibited the growth of *E. coli*. [29]. In addition, the acidity of citric acid could affect the metabolism of bacteria by changing the intracell pH. However, composites of CNFs with PVA showed negligible inhibition for both strains. This result could be ascribed to the fact that the dosage of CNFs (5%) was under the threshold value of antibacterial inhibition.

## 4. Conclusions

Citric acid acts as a reactant for the esterification of cellulose and a hydrogen bond donor in a deep eutectic solvent for the fibrillation of cellulose. Increasing the pretreatment temperature and prolonging the duration triggered the degradation of cellulose and further promoted esterification, resulting in a small size and the high DS of CNFs. With the increase in carboxyl groups on the cellulosic backbone, CNFs showed a favorable anti-bacterial activate for *E. coli* and *S. aureus*. Composites of CNFs at a dosage of 5 wt% enhanced the mechanical properties of the PVA film. Compared with the CNF samples, CNF obtained from treatment at 120 °C for 24 h with considerable size and DS was the optimum supplement. However, the composite films exhibited negligible anti-bacterial activate compared to CNFs.

## Figures and Tables

**Figure 1 polymers-15-00148-f001:**
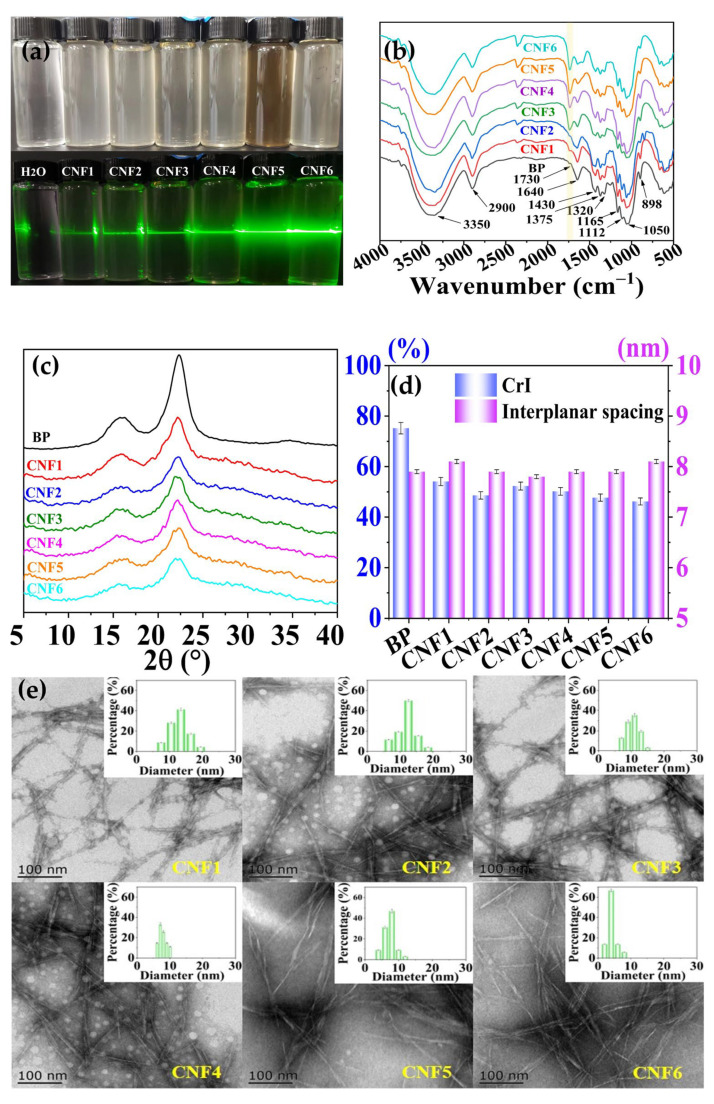
Structural characteristics of the CNFs. (**a**) Tyndall effect of CNFs; (**b**) FTIR spectra of BP and CNFs; (**c**) XRD diffraction of the BP and CNFs; (**d**) crystalline and interplanar spacing of the BP and CNFs; (**e**) TEM and the diameter distribution of the CNFs.

**Figure 2 polymers-15-00148-f002:**
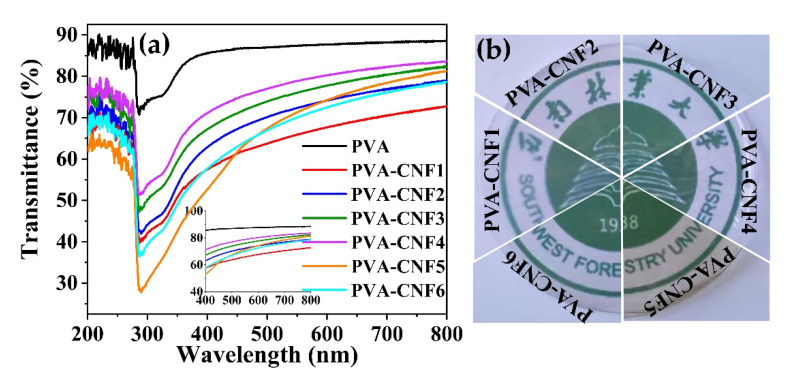
Transmittance of the pure PVA and composite films (**a**): transmittance of films; (**b**): visibility of films.

**Figure 3 polymers-15-00148-f003:**
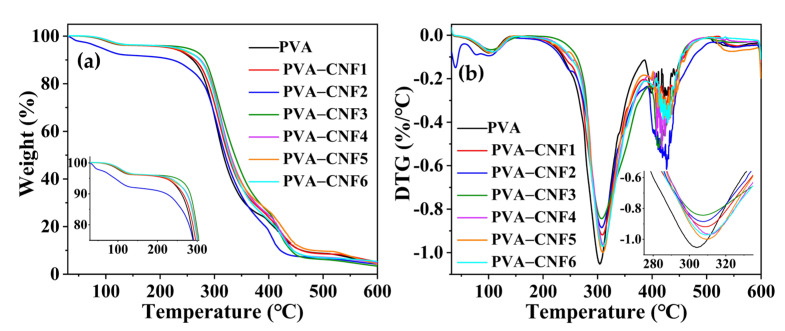
The TG (**a**) and DTG (**b**) curves of the pure PVA and composite films.

**Figure 4 polymers-15-00148-f004:**
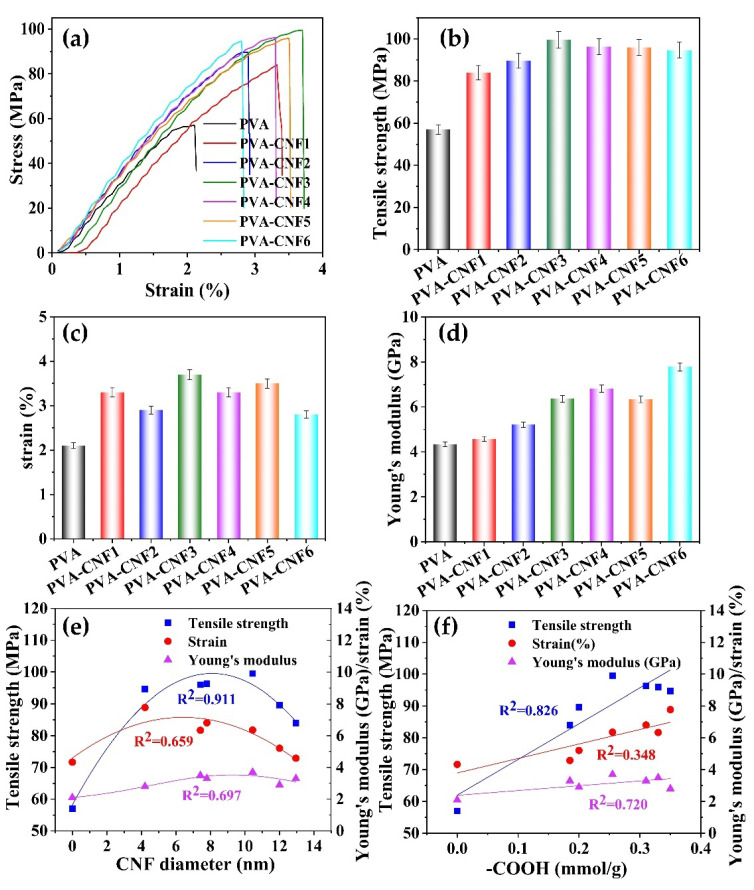
Mechanical properties of the pure PVA and composite films. (**a**) Stress; (**b**) tensile strength; (**c**) strain; (**d**) Young’s modulus; (**e**) relationship between the CNF diameter and mechanical properties of the composite films; (**f**) relationship between the carboxyl group content and mechanical properties of the composite films.

**Figure 5 polymers-15-00148-f005:**
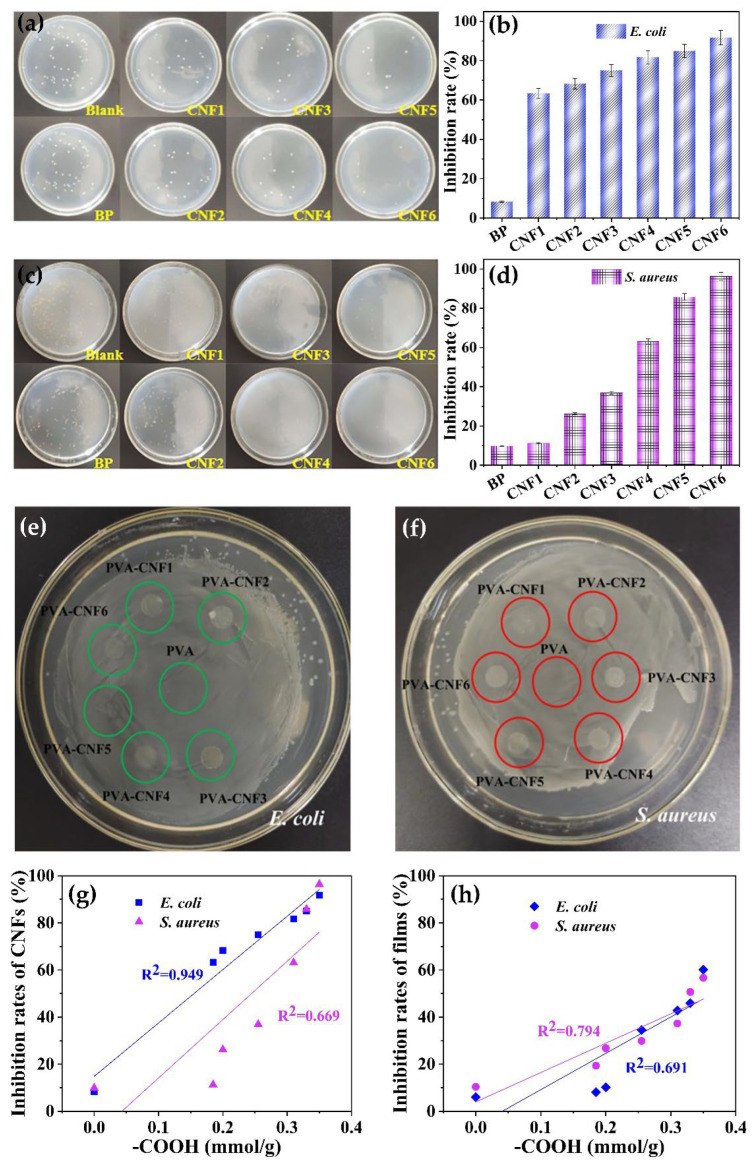
Inhibition of CNFs for *E. coli* (**a**,**b**) and *S. aureus* (**c**,**d**), inhibition of composite films for *E. coli* (**e**) and *S. aureus* (**f**), and the relationship between the carboxylic group content and the inhibition of CNF (**g**) and the composite films (**h**).

**Table 1 polymers-15-00148-t001:** Yields, carboxyl group content, and zeta potential of CNF samples.

CNFs	Temperature (°C)	Time (h)	Yield (%)	−COOH (mmol/g)	Zeta Potential (mV)
BP				Non detectable	Non detectable
CNF1	100	12	84.5	0.19	−6.24
CNF2	100	24	84.1	0.20	−7.24
CNF3	120	12	81.2	0.26	−14.1
CNF4	120	24	76.0	0.31	−16.9
CNF5	140	12	71.2	0.33	−16.1
CNF6	140	24	66.6	0.35	−16.63

## Data Availability

The data supporting this study’s findings are available from the corresponding author upon reasonable request.

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
