# Peer review of "Preparation of Anti-Bacterial Cellulose Nanofibrils (CNFs) from Bamboo Pulp in a Reactable Citric Acid–Choline Chloride Deep Eutectic Solvent"

_polymers, 2022, doi:10.3390/polym15010148_

Round 1
Reviewer 1 Report
Please see the attched.

Author Response
The language has been improved, the changes are marked in red in the manuscript attached below

Reviewer 2 Report
The manuscript is well written, and the author tries to claim the anti-bacterial properties of extracted cellulose nanofibrils (CNFs) from bamboo pulp.
1. The figure's quality is poor, and it's hard to read the axis title.
2. The author can provide storage and loss modulus to see the differences
Author Response
Comments and Suggestions for Authors
The manuscript is well written, and the author tries to claim the anti-bacterial properties of extracted cellulose nanofibrils (CNFs) from bamboo pulp.
- The figure's quality is poor, and it's hard to read the axis title.
Response: The resolution of figures has been improved.
- The author can provide storage and loss modulus to see the differences
Response: Yes! Dynamic mechanical analyzer is a good method to analyze the mechanical properties of materials. Thank you very much for the kind suggestion for us. Previous research has suggested that CNC should be modified or adding a crosslinking agent to effectively enhance mechanical properties of CNC-PVA composite film. In our study, CNFs with 0.19-0.35 mmol/g was prepared by DES treatment. The large molecular and low degree of substitution of citric acid made it difficult to form covalent bond between carboxyl groups of CNFs and hydroxyl groups of PVA due to steric hindrance. Thus, the composite films were not submitted to dynamic mechanical analysis. In further study, we will analyze mechanical properties of materials in detail.
Tanpichai, S.; Oksman, K. Crosslinked poly(vinyl alcohol) composite films with cellulose nanocrystals: Mechanical and thermal properties. J. Appl. Polym. Sci. 2018, 135, 45710, https://doi.org/10.1002/app.45710
All changes are marked in the revised manuscript that is attached below

Reviewer 3 Report
Manuscript was well written, synthesis of CNFs from bamboo pulp is a very good approach, however the bamboos availability for the CNFs preparation is subject to environmental degradation. Besides, the synthesis and characterization of CNFs with its anti bacterial application is well studied. Authors need to explain the selection of dosage as 5 Wt % of PVA composite films.
Author Response
Comments and Suggestions for Authors
Manuscript was well written, synthesis of CNFs from bamboo pulp is a very good approach, however the bamboos availability for the CNFs preparation is subject to environmental degradation. Besides, the synthesis and characterization of CNFs with its anti bacterial application is well studied. Authors need to explain the selection of dosage as 5 Wt % of PVA composite films.
Response: Yes! The bamboos availability for the CNFs preparation is subject to environmental degradation. However, bamboo is one of the most important lignocellulosic biomass in Yunnan Province, China. Yunnan province has 340,000 ha groves of bamboo, and containing sympodial bamboo species as the main species. Beside, pulping and paper making is the most important bamboo processing and utilization technologies, producing about 100,000 t bamboo pulp per year. Thus, bamboo pulp was used to prepare CNF in this study.
The dosage of CNF was selected according to our previous research. In which, effect of CNF dosage (1, 5, and 10 wt%) on characteristics of PVA films was comparatively studied. These results suggested that PVA film containing 5 wt% CNF had favorable mechanical properties. However, increasing the CNF content to 10% introduced a decrease in mechanical properties of composite films. Thus, 5 wt% CNF dosage was used in this study.
Zhu, Y.C.; Zhang, J.H.; Zhao, P.; Wang, D.W.; Shi, Z.J.; Yang, J.; Yang, H.Y. Fabrication of cellulose nanocrystals (CNCs) in choline chloride-citric acid (ChCl-CA) solvent to lodge antimicrobial activity. Bioresource. 2022, 17(3), 4347–4359, http://doi.org/10.15376/biores.17.3.4347-4359
Tanpichai, S.; Oksman, K. Crosslinked poly(vinyl alcohol) composite films with cellulose nanocrystals: Mechanical and thermal properties. J. Appl. Polym. Sci. 2018, 135, 45710, https://doi.org/10.1002/app.45710

Reviewer 4 Report
Zhu et al. reported a method for fibrillating and esterifying bamboo pulp in cellulose nanofibrils (CNF) in one-pot citric acid choline chloride deep eutectic solvent treatment with various conditions and compositing with PVA. The resulting products demonstrated improved anti-bacterial activities against E. coli and S. aureus. This manuscript is inspiring and useful. I recommend the following issues be addressed:
1. Why is the solution color of CNF5 so different from other CNF but CNF5 is chemically similar to those of others?
2. How did the author quantify the interplanar spacing?
3. Does the TEM samples of CNF require negative staining?
4. What does the inlet figure of 2a represent?
5. All figures’resolution is not high enough and graphical texts are too small
6. Figure 5 requires some statistical analysis
7. I don’t see the colony of bacterial in each island show significant anti-bacterial activity in figure 5. It looks like bacteria is more concentrated on different PVA-CNF ratio.
8. Some applications of other PVA-based transparent film can be discussed to compare with cellulalose PVA composite (e.g., citrate-based fluorophores in PVA matrix: Journal of Materials Science 2019, 54 (2), 1236-1247)
9. How was the % of COOH group characterized?
10. The English language of this manuscript should be further polished.
Author Response
Comments and Suggestions for Authors
Zhu et al. reported a method for fibrillating and esterifying bamboo pulp in cellulose nanofibrils (CNF) in one-pot citric acid choline chloride deep eutectic solvent treatment with various conditions and compositing with PVA. The resulting products demonstrated improved anti-bacterial activities against E. coli and S. aureus. This manuscript is inspiring and useful. I recommend the following issues be addressed:
- Why is the solution color of CNF5 so different from other CNF but CNF5 is chemically similar to those of others?
Response: The dark suspension of CNF5 might be ascribed the fact that hemicellulose and partial amorphous cellulose in the secondary cell wall was also degraded at higher temperatures (140 ºC). After reaction, water was used at as anti-solvent and added into the mixture, leading to deposition of the degradation products (such as furfural, a deep brown products) onto the CNFs. This phenomenon has also been observed by Liu et al., (2017). However, the light color of suspension CNF6 as compared with CNF5 might be due to the fat that the degradation products were partly evaporated during long duration. The discussion has been added and marked in blue in the manuscript.
Liu, Y.Z.; Guo, B.T.; Xia, Q.Q.; Meng, J.; Chen, W.S.; Liu, S.X.; Wang, Q.W.; Liu, Y.X.; Li, J.; Yu, H.P. Efficient cleavage of strong hydrogen bonds in cotton by deep eutectic solvents and facile fabrication of cellulose nanocrystals in high yields. ACS Sustainable Chem. Eng. 2017, 5, 9, 7623–7631, https://doi.org/10.1021/acssuschemeng.7b00954
- How did the author quantify the interplanar spacing?
Response: The detail for interplanar spacing calculation was conducted according to a literature as following. It also has been added and marked in blue in the manuscript.
The interplanar spacing of CNF was calculated on the software according to a literature as following equation:
Where λ is the X-ray wavelength, θ is the scattering angle, β is the half width of diffraction peak.
Ling, Z.; Guo, Z.; Huang, C.X.; Yao, L.; Xu, F. Deconstruction of oriented crystalline cellulose by novel levulinic acid based deep eutectic solvents pretreatment for improved enzymatic accessibility. Bioresour. Technol. 2020, 123025, https://doi.org/10.1016/j.biortech.2020.123025
- Does the TEM samples of CNF require negative staining?
Response: The CNF was in nano size and difficult to observe without staining. During TEM analysis of organic compounds, phosphotungstic acid is usually used to negative staining. Phosphotungstic acid is heavier than organic compound and has better electrons blocking ability, forming a black background. Thus, the CNF emerges as bright and can be observed easily.
- What does the inlet figure of 2a represent?
Response: The inlet figure of 2a is the transmittance of films in wavelength range of 400-800 nm. Generally, the light at about 550 nm is sensitive to human vision. Thus, the transmittance spectra of films in wavelength range of 400-800 nm were enlarged.
- All figures’ resolution is not high enough and graphical texts are too small.
Response: Yes! All the figures have been improved.
- Figure 5 requires some statistical analysis.
Response: Statistical analysis has been added in the manuscript and marked in blue.
- I don’t see the colony of bacterial in each island show significant anti-bacterial activity in figure 5. It looks like bacteria is more concentrated on different PVA-CNF ratio.
Response: Yes, increasing the CNF ratio could improve the anti-bacterial activity of composite films. However, large dosage of CNF might introduce decreased of mechanical properties.
Zhu, Y.C.; Zhang, J.H.; Zhao, P.; Wang, D.W.; Shi, Z.J.; Yang, J.; Yang, H.Y. Fabrication of cellulose nanocrystals (CNCs) in choline chloride-citric acid (ChCl-CA) solvent to lodge antimicrobial activity. Bioresource. 2022, 17(3), 4347–4359, http://doi.org/10.15376/biores.17.3.4347-4359
Tanpichai, S.; Oksman, K. Crosslinked poly(vinyl alcohol) composite films with cellulose nanocrystals: Mechanical and thermal properties. J. Appl. Polym. Sci. 2018, 135, 45710, https://doi.org/10.1002/app.45710
- Some applications of other PVA-based transparent film can be discussed to compare with cellulose PVA composite (e.g., citrate-based fluorophores in PVA matrix: Journal of Materials Science 2019, 54 (2), 1236-1247)
Response: The optical properties of CNF-PVA films were comparatively discussed with the citrate-based fluorophores in PVA matrix. The changes have been marked in blue in the manuscript.
- How was the % of COOH group characterized?
Response: The carboxyl group content in CNFs was determined by a NaOH titration method according to a literature [11]. In detail, 30 mL 1% CNF suspension, 0.5 mL NaCl (1 mM) and 0.5 mL HCl (0.1 M) were added into a beaker and stirred for 30 min. Afterwards, the mixture was titrated with 0.01 M NaOH aqueous solution. The content of carboxyl group was determined based on the following equation:
where V1 and V2 represent the volumes of NaOH (mL) solution at the beginning and end positions of the turning point in the conductimetric titration curves, C represents the concentration of NaOH solution (0.01 M), w represents the absolute dry weight of the CNFs (g).
The detail for COOH group characterization has been added and marked in blue in the manuscript.
- The English language of this manuscript should be further polished.
Response: Yes, the English languages of this manuscript has been carefully revised and the changes are marked in red.

Round 2
Reviewer 1 Report
NA
Reviewer 4 Report
The authors have addressed my comments.